# Some simulations of age-period-cohort analysis applying Bayesian regularization: Conditions for using random walk model

Yuta Matsumoto [ORCID]*

Quality Assurance Office Institutional Research, Hosei University, Chiyoda-ku, Tokyo, Japan

* yuta.matsumoto.76@adm.hosei.ac.jp

**Data availability statement:** All relevant data are within the manuscript and its Supporting information files.

## Abstract

Age-period-cohort (APC) analysis, one of the fundamental time-series models, has an identification problem of the inability to separate linear components of the three effects. However, constraints to solve the problem are still controversial because multilevel analysis used in many studies results in the linear component of cohort effects being close to zero. In addition, previous studies do not compare the Bayesian cohort model proposed by Nakamura with the well-known intrinsic estimator. This paper focuses on three models of Bayesian regularization using priors of normal distributions. A random effects model refers to multilevel analysis, a ridge regression model is equivalent to the intrinsic estimator, and a random walk model refers to the Bayesian cohort model. Here, applying Bayesian regularization in APC analysis is to estimate linear components by using nonlinear components and priors. We aim to suggest conditions for using the random walk model by comparing the three models through some simulations with settings for the linear and nonlinear components. Simulation 1 emphasizes an impact of the indexes by making absolute values of the nonlinear components small. Simulation 2 randomly generates the amounts of change in the linear and nonlinear components. Simulation 3 randomly generates artificial parameters with only linear components are less likely to appear, to consider the Bayesian regularization assumption. As a result, Simulation 1 shows the random walk model, unlike the other two models, mitigates underestimating the linear component of cohort effects. On the other hand, in Simulation 2, none of the models can recover the artificial parameters. Finally, Simulation 3 shows the random walk model has less bias than the other models. Therefore, there is no one-size-fits-all APC analysis. However, this paper suggests the random walk model performs relatively well in data generating processes, where only linear components are unlikely to appear.

## Introduction

Age-period-cohort (APC) analysis is one of the fundamental time-series analyses used in many research fields. In APC analysis, age effects reflect the influence of individual differences in age and period effects reflect the influence of differences in time period. Cohort effects represent the influence of differences in birth year. APC analysis is important because long-term

**Funding:** The author(s) received no specific funding for this work.

**Competing interests:** The author has declared that no competing interests exist.

changes are also the result of demographic metabolism, in which older generations leave society and younger generations with different characteristics enter [1]. Given the nature of such cohort replacement, we need to consider not only period effects but also cohort effects.

It is well-known that APC analysis has a serious issue of identification. In general, cohort is linearly associated with age and period according to the relationship cohort = period – age. This linear dependence of the three factors confounds linear components of the three effects. In other words, the APC identification problem makes it impossible to directly estimate the linear components of these effects. There are various constraints, such as using unequal-interval widths for age, period, and cohort indexes [2]. Many previous studies have applied multilevel analysis to solve the rank deficiency of the design matrix. However, this constraint is problematic [3], especially because it results in the linear component of cohort effects being close to zero [4,5]. The intrinsic estimator [6] is also a well-known method in APC analysis. This method is sensitive to the type of dummy parameterization of the design matrix [7,8], and is difficult to verify in empirical research [9].

On the other hand, there are few studies using the Bayesian cohort model proposed by Nakamura [10]. Sakaguchi and Nakamura [11] suggest that this assumption, unlike multilevel analysis, mitigates underestimating the linear component of cohort effects caused by the indexes of the three effects. However, these constraints are still controversial [12], as the above previous studies do not compare the model with the intrinsic estimator and do not evaluate the performance of the three models. Furthermore, they do not show whether the mitigation of underestimating the linear component of cohort effects reduces the bias in APC analysis. Therefore, it is unclear when we should use the Bayesian cohort model.

This paper examines the major models of APC analysis by considering some simulations with settings for the linear and nonlinear components. To compare the assumed constraints, we focus on three models of Bayesian regularization using prior probabilities of normal distributions. A random effects model refers to multilevel analysis, a ridge regression model is equivalent to the intrinsic estimator, and a random walk model refers to the Bayesian cohort model. Here, the linear components of the three effects are estimated using the nonlinear components and the priors. This paper evaluates the three models in terms of how well the linear components are recovered. The first simulation makes absolute values of the nonlinear components small to emphasize an impact of the indexes mentioned in the previous studies. The next simulation randomly generates the amounts of change in linear and nonlinear components according to normal distributions. The following simulation sets artificial parameters so that a pattern with only linear components is unlikely to appear, to consider the Bayesian regularization assumption. By comparing the three models through these simulations, this paper suggests conditions for using the random walk model.

This paper reviews APC analysis in "Theory" section. Specifically, "APC analysis" subsection shows the notation and the identification problem. "Bayesian regularization" subsection describes the constraints for the random effects model, the ridge regression model, and the random walk model. "Mathematical mechanism" subsection shares the why the previous study suggests that the random walk model performs better than the random effects model. "Methods" section presents systematic simulations adopted in this paper and the definition of a bias evaluation function. "Results" section verifies the performance of the three models through these simulations. "Discussion" section concludes that while there is no one-size-fits-all APC analysis, the random walk model performs relatively well in data generating processes, where only linear components are unlikely to appear.

## Theory

### APC analysis

**Notation.** Let $i = 1, \ldots, I$ denote the index of the age group, $j = 1, \ldots, J$ denote the index of the period group, and $k = 1, \ldots, K$ denote the index of the cohort group. These three indexes be determined by

$$k = j - i + I, \quad K = I + J - 1, \tag{1}$$

if the intervals of age and period have the same scale. The general model for APC analysis is

$$y_{i,j} = b_0 + b_i^A + b_j^P + b_k^C + \epsilon_{i,j}, \tag{2}$$

where $y_{i,j}$ denotes the observed value (see Table 1), $b_0$ denotes the intercept, $b_i^A$ denotes the age effect, $b_j^P$ denotes the period effect, $b_k^C$ denotes the cohort effect, $\epsilon_{i,j}$ denotes the error term, and each effect satisfies the sum-to-zero condition,

$$\sum_{i=1}^{I} b_i^A = \sum_{j=1}^{J} b_j^P = \sum_{k=1}^{K} b_k^C = 0.$$

Here, when we approximate the error terms with normal distributions, the model becomes

$$y_n \sim \text{Normal}\,(\mu_n, \sigma) \qquad n = 1, \ldots, N,$$

$$\mu_n = b_0 + \sum_{i=1}^{I} x_{n,i}^A b_i^A + \sum_{j=1}^{J} x_{n,j}^P b_j^P + \sum_{k=1}^{K} x_{n,k}^C b_k^C,$$

where $y_n$ is $y_{i,j}$ rearranged as the component of a vector with $N$ rows, $\sigma$ denotes the standard deviation, and $x_{n,i}^A$, $x_{n,j}^P$, and $x_{n,k}^C$ are the components of the design matrix composed of three factors. Then, the log likelihood is

$$\log L = -N \log \sigma - \frac{1}{2\sigma^2} \sum_{n=1}^{N} (y_n - \mu_n)^2, \tag{3}$$

excluding the constant term. In general, estimates are obtained by maximizing Eq (3); however, we need to add constraints in APC analysis since it is not possible to uniquely determine the estimates owing to the identification problem described below.

**Identification problem.** To understand the identification problem, it is convenient to center each index [13],

$$v_i^A = i - \frac{I+1}{2}, \quad v_j^P = j - \frac{J+1}{2}, \quad v_k^C = k - \frac{K+1}{2}.$$

**Table 1. Observed values in an age-period table.**

| $y_{i,j}$, $k$ | $j = 1$ | $\cdots$ | $j = J$ |
|---|---|---|---|
| $i = 1$ | $y_{1,1}$, $k = I$ | $\cdots$ | $y_{1,J}$, $k = K$ |
| $\vdots$ | $\vdots$ | $\ddots$ | $\vdots$ |
| $i = I$ | $y_{I,1}$, $k = 1$ | $\cdots$ | $y_{I,J}$, $k = J$ |

Here, the equation

$$v_i^A - v_j^P + v_k^C = 0,$$

is satisfied using the relationship of the cohort index in Eq (1). Thus, the right-hand side of Eq (2) becomes

$$b_0 + b_i^A + b_j^P + b_k^C + \epsilon_{i,j} = b_0 + b_i^A + b_j^P + b_k^C + \left(v_i^A - v_j^P + v_k^C\right) + \epsilon_{i,j},$$

and we can write the general solutions of the three effects as

$$b_i^A = \widehat{b_i^A} + sv_i^A, \quad b_j^P = \widehat{b_j^P} - sv_j^P, \quad b_k^C = \widehat{b_k^C} + sv_k^C, \tag{4}$$

where $\widehat{b_i^A}$, $\widehat{b_j^P}$, and $\widehat{b_k^C}$ are the particular solutions of the three effects and $s$ denotes an arbitrary real number.

In summary, the APC identification problem is that there are many maximum likelihood estimates of Eq (3) owing to the linear dependency of cohort = period – age. In other words, the linear components of the three effects affected by $v_i^A$, $v_j^P$, and $v_k^C$ cancel each other out completely when the slopes of the age and cohort effects increase by $s$ and the slope of the period effect decreases by $s$. On the other hand, we can easily separate the nonlinear components that are irrelevant to the identification problem.

## Bayesian regularization

To overcome the identification problem, Bayesian regularization constrains parameters of the three effects by assuming prior probabilities. It is a strategy to statistically estimate mathematically indistinguishable linear components by using mathematically identifiable nonlinear components and priors. Here, if there are an infinite number of maximum likelihood estimates, as in APC analysis, point estimates that maximize the posterior probabilities of Bayesian models are determined by maximizing the priors.

**Random effects model.** Many studies use the multilevel analysis that reflects the nesting of individuals in groups of period and cohort [14]. They treat the age effects as fixed effects, which means that the age effects are unconstrained. In this paper, we consider the random effects model with reference to multilevel analysis, where the model assumes a normal distribution for the prior probabilities of each of the three effects. The priors are

$$\begin{aligned} b_i^A &\sim \text{Normal}\left(0, \sigma^A\right) & i &= 1, \dots, I, \\ b_j^P &\sim \text{Normal}\left(0, \sigma^P\right) & j &= 1, \dots, J, \\ b_k^C &\sim \text{Normal}\left(0, \sigma^C\right) & k &= 1, \dots, K, \end{aligned}$$

where $\sigma^A$, $\sigma^P$, and $\sigma^C$ denote the standard deviations of the three effects. The log priors are

$$\begin{aligned} \log RE = &- \left(I \log \sigma^A + J \log \sigma^P + K \log \sigma^C\right) \\ &- \frac{1}{2}\left\{\frac{1}{(\sigma^A)^2}\sum_{i=1}^{I}(b_i^A)^2 + \frac{1}{(\sigma^P)^2}\sum_{j=1}^{J}(b_j^P)^2 + \frac{1}{(\sigma^C)^2}\sum_{k=1}^{K}(b_k^C)^2\right\}, \end{aligned} \tag{5}$$

excluding the constant term. Here, maximizing Eq (5) means minimizing the sum of squares of the parameters.

**Ridge regression model.** Ridge regression analysis is a method that imposes the sum of squares of parameters as a penalty. The aim is to overcome the adverse effects of multicollinearity. The ridge regression model is implemented by assuming normal distributions with zero means and equal standard deviations for the prior probabilities of the three effects. Unifying the standard deviations,

$$\lambda = \sigma^A = \sigma^P = \sigma^C, \tag{6}$$

and substituting Eq (6) into Eq (5), we can write the log priors as

$$\log RR = -(I + J + K)\log\lambda - \frac{1}{2\lambda^2}\left\{\sum_{i=1}^{I}(b_i^A)^2 + \sum_{j=1}^{J}(b_j^P)^2 + \sum_{k=1}^{K}(b_k^C)^2\right\}. \tag{7}$$

Here, maximizing Eq (7) means minimizing the sum of squares of the parameters as well as $\log RE$.

The intrinsic estimator is another well-known method in APC analysis and produces similar results to the ridge regression model. The reason is that this operation minimizes the Euclidean norm of the parameters, giving a particular solution that is the average of the general solution [15].

**Random walk model.** We can also apply time-series models to APC analysis based on the previous study that proposes smoothing cohort effects [16]. The random walk model literally assumes a random walk for the prior probabilities of the three effects [17]. We write this model as

$$b_{i+1}^A \sim \text{Normal}\,(b_i^A, \sigma^A) \qquad\qquad i = 1, \dots, I - 1,$$
$$b_{j+1}^P \sim \text{Normal}\,(b_j^P, \sigma^P) \qquad\qquad j = 1, \dots, J - 1,$$
$$b_{k+1}^C \sim \text{Normal}\,(b_k^C, \sigma^C) \qquad\qquad k = 1, \dots, K - 1.$$

The log priors can be summarized as follows:

$$\log RW = -\left\{(I-1)\log\sigma^A + (J-1)\log\sigma^P + (K-1)\log\sigma^C\right\}$$
$$-\frac{1}{2}\left\{\frac{1}{(\sigma^A)^2}\sum_{i=1}^{I-1}(b_{i+1}^A - b_i^A)^2 + \frac{1}{(\sigma^P)^2}\sum_{j=1}^{J-1}(b_{j+1}^P - b_j^P)^2 + \frac{1}{(\sigma^C)^2}\sum_{k=1}^{K-1}(b_{k+1}^C - b_k^C)^2\right\}, \tag{8}$$

excluding the constant term.

Here, maximizing Eq (8) means, unlike $\log RE$ and $\log RR$, minimizing the sum of squares of the differences in the adjacent parameters. Furthermore, the random walk model is equivalent to the Bayesian cohort model proposed by Nakamura [10] and this constraint takes advantage of the fact that age, period, and cohort indexes are ordered.

## Mathematical mechanism

**Linear and nonlinear components.** APC analysis depends heavily on the way in which the constraints assign the linear components to the three effects. Thus, this paper separates the linear and nonlinear components of the general solution as $b_i^A = b_i^{A[L]} + b_i^{A[NL]}$ [11] in order to discuss constraint bias. For example, we regress the particular solution of the age effects on the centering index and $s^A$ denotes the obtained slope. The equation is $\widehat{b}_i^A = s^A v_i^A + \epsilon_i^A$ and $\epsilon_i^A = b_i^{A[NL]}$, as $\epsilon_i^A$ does not contain the linear component. Here, the particular solutions of the

three effects are

$$\widehat{b}_i^A = s^A v_i^A + b_i^{A[NL]}, \quad \widehat{b}_j^P = s^P v_j^P + b_j^{P[NL]}, \quad \widehat{b}_k^C = s^C v_k^C + b_k^{C[NL]},$$

where $s^P$ and $s^C$ are the slopes calculated from the particular solutions of the period and cohort effects. By substituting the above solutions into Eq (4), we rewrite the general solutions as follows:

$$b_i^A = (s^A + s) v_i^A + b_i^{A[NL]}, \quad b_j^P = (s^P - s) v_j^P + b_j^{P[NL]},$$
$$b_k^C = (s^C + s) v_k^C + b_k^{C[NL]}. \tag{9}$$

Therefore, the linear components of the general solutions are

$$b_i^{A[L]} = (s^A + s) v_i^A, \quad b_j^{P[L]} = (s^P - s) v_j^P, \quad b_k^{C[L]} = (s^C + s) v_k^C, \tag{10}$$

using the centering indexes.

**Linear components represented by indexes.**  We express the linear components of the general solutions for the models of Bayesian regularization using the centering indexes. Here, the equation

$$\sum_{i=1}^I b_i^{A[L]} b_i^{A[NL]} = (s^A + s) \sum_{i=1}^I v_i^A b_i^{A[NL]} = 0,$$

is satisfied because the linear and nonlinear components are orthogonal. The sum of squares of the parameters is

$$\sum_{i=1}^I (b_i^A)^2 = \sum_{i=1}^I \left\{ (s^A + s) v_i^A + b_i^{A[NL]} \right\}^2 = (s^A + s)^2 \sum_{i=1}^I (v_i^A)^2 + \sum_{i=1}^I (b_i^{A[NL]})^2,$$

using Eq (9). Therefore, the log priors of the random effects model in Eq (5) become

$$\log RE^A = -I \log \sigma^A - \frac{1}{2(\sigma^A)^2} \left\{ (s^A + s)^2 \sum_{i=1}^I (v_i^A)^2 + \sum_{i=1}^I (b_i^{A[NL]})^2 \right\},$$

$$\log RE^P = -J \log \sigma^P - \frac{1}{2(\sigma^P)^2} \left\{ (s^P - s)^2 \sum_{j=1}^J (v_j^P)^2 + \sum_{j=1}^J (b_j^{P[NL]})^2 \right\},$$

$$\log RE^C = -K \log \sigma^C - \frac{1}{2(\sigma^C)^2} \left\{ (s^C + s)^2 \sum_{k=1}^K (v_k^C)^2 + \sum_{k=1}^K (b_k^{C[NL]})^2 \right\},$$

$$\log RE = \log RE^A + \log RE^P + \log RE^C. \tag{11}$$

We obtain the log priors of the ridge regression model by substituting Eq (6) into Eq (11). Next, the sum of squares of the differences in the adjacent parameters is

$$\sum_{i=1}^{I-1} (b_{i+1}^A - b_i^A)^2 = \sum_{i=1}^{I-1} \left\{ (s^A + s) + (b_{i+1}^{A[NL]} - b_i^{A[NL]}) \right\}^2$$

$$= (s^A + s)^2 (I - 1) + 2(s^A + s)(b_I^{A[NL]} - b_1^{A[NL]}) + \sum_{i=1}^{I-1} (b_{i+1}^{A[NL]} - b_i^{A[NL]})^2,$$

using the general solutions of Eq (9). Thus, the log priors of the random walk model in Eq (8) become

$$
\log RW^A = -(I-1)\log\sigma^A - \frac{1}{2(\sigma^A)^2}\Bigg\{(s^A+s)^2(I-1)
$$

$$
+ 2(s^A+s)(b_I^{A[NL]} - b_1^{A[NL]}) + \sum_{i=1}^{I-1}(b_{i+1}^{A[NL]} - b_i^{A[NL]})^2\Bigg\},
$$

$$
\log RW^P = -(J-1)\log\sigma^P - \frac{1}{2(\sigma^P)^2}\Bigg\{(s^P-s)^2(J-1)
$$

$$
+ 2(s^P-s)(b_J^{P[NL]} - b_1^{P[NL]}) + \sum_{j=1}^{J-1}(b_{j+1}^{P[NL]} - b_j^{P[NL]})^2\Bigg\},
$$

$$
\log RW^C = -(K-1)\log\sigma^C - \frac{1}{2(\sigma^C)^2}\Bigg\{(s^C+s)^2(K-1)
$$

$$
+ 2(s^C+s)(b_K^{C[NL]} - b_1^{C[NL]}) + \sum_{k=1}^{K-1}(b_{k+1}^{C[NL]} - b_k^{C[NL]})^2\Bigg\},
$$

$$
\log RW = \log RW^A + \log RW^P + \log RW^C. \tag{12}
$$

**Index weights of linear components.** The linear components cause the difference in the estimates and are weighted by the indexes in the general solutions of Eq (10). Focusing on $(s^A+s)^2$, $(s^P-s)^2$, and $(s^C+s)^2$ that are common to these models, the terms in the random effects model of Eq (11) are

$$
(s^A+s)^2\sum_{i=1}^{I}(v_i^A)^2, \quad (s^P-s)^2\sum_{j=1}^{J}(v_j^P)^2, \quad (s^C+s)^2\sum_{k=1}^{K}(v_k^C)^2. \tag{13}
$$

Here, maximizing $\log RE$ often satisfies $s$ where $(s^C+s)^2$ is smaller than $(s^A+s)^2$ and $(s^P-s)^2$ because $K=I+J-1$ makes the index weights $\sum_{k=1}^{K}(v_k^C)^2 > \sum_{i=1}^{I}(v_i^A)^2$ and $\sum_{k=1}^{K}(v_k^C)^2 > \sum_{j=1}^{J}(v_j^P)^2$. In other words, the index weights exert a strong pressure to shrink the linear component of the cohort effects; consequently, they tend to be flat. The above also occurs with the ridge regression model. The terms in the random effects model of Eq (12) are

$$
(s^A+s)^2(I-1), \quad (s^P-s)^2(J-1), \quad (s^C+s)^2(K-1). \tag{14}
$$

For the same reason, it also tends to underestimate the linear component of the cohort effects owing to the index.

Bayesian regularizations using normal distributions have in common that the linear component of the cohort effects is close to zero. Here, Sakaguchi and Nakamura [11] suggested that the index weights of the random effects model had a greater impact than those of the random walk model. However, the previous study did not verify that the index weights of the random effects model were large regardless of the values of $I$ or $J$. Consequently, this paper compares the impact of the index weights in Eqs (13) and (14). Focusing on period and cohort, the ratio is $J-1:K-1$ for the random walk model and $\sum_{j=1}^{J}(v_j^P)^2:\sum_{k=1}^{K}(v_k^C)^2$ for the

 

other two models. The squared sums of the centering index are

$$\sum_{j=1}^{J}(v_j^P)^2 = \frac{1}{12}J(J+1)(J-1), \quad \sum_{k=1}^{K}(v_k^C)^2 = \frac{1}{12}K(K+1)(K-1).$$

Here, a comparison of the above ratios of the index weights shows

$$\frac{\sum_{k=1}^{K}(v_k^C)^2}{\sum_{j=1}^{J}(v_j^P)^2} - \frac{K-1}{J-1} = \frac{(K-1)(2J+I)(I-1)}{J(J+1)(J-1)} > 0. \tag{15}$$

Thus, the comparison of the index weights in Eq (15) suggests that the random walk model is less affected by the index weights than are the random effects and ridge regression models. In other words, minimizing the sum of squares of the differences in the adjacent parameters rather than the parameters themselves mitigates underestimating the linear component of cohort effects.

## Methods

### Artificial parameter and bias evaluation function

This paper examines some simulations to evaluate the performance of the three models applying Bayesian regularization. This subsection describes a common framework for the simulations. Since estimation of the linear components is important for the identification problem, artificial parameters need to be separated into linear and nonlinear components. There are some candidates for data generating processes, including polynomial functions. However, analysts cannot freely determine each component of artificial parameters generated by polynomial functions, because they contain not only linear components but also nonlinear components. Thus, this paper uses trigonometric functions for the nonlinear components, as they contain no linear components and can be set to any amount of change. Another reason is to demonstrate the robustness of the results by adopting data generating processes that do not match the priors of normal distributions. Accordingly, the artificial parameters of the three effects are as follows:

$$\beta_i^A = \beta_0^A + \beta^{A[L]}v_i^A + \beta^{A[NL]}\cos(\pi i),$$
$$\beta_j^P = \beta_0^P + \beta^{P[L]}v_j^P + \beta^{P[NL]}\cos(\pi j),$$
$$\beta_k^C = \beta_0^C + \beta^{C[L]}v_k^C + \beta^{C[NL]}\cos(\pi k),$$

where $\beta_i^{A[L]}$, $\beta_j^{P[L]}$, and $\beta_k^{C[L]}$ denote the slopes of the artificial parameters, $\beta_i^{A[NL]}$, $\beta_j^{P[NL]}$, and $\beta_k^{C[NL]}$ denote the amounts of change in the nonlinear components. Since $\sum_{i=1}^{I}\beta^{A[NL]}\cos(\pi i) = -\beta^{A[NL]}$ when $I$ is an odd number, adding $-\frac{\beta^{A[NL]}}{2I}\{\cos(\pi I)-1\}$ to each artificial parameter of the age effect satisfies $\sum_{i=1}^{I}\beta_i^A = 0$. Therefore, the zero-sum condition of $\sum_{i=1}^{I}\beta_i^A = \sum_{j=1}^{J}\beta_j^P = \sum_{k=1}^{K}\beta_k^C = 0$ are derived by setting $\beta_0^A = -\frac{\beta^{A[NL]}}{2I}\{\cos(\pi I)-1\}$, $\beta_0^P = -\frac{\beta^{P[NL]}}{2J}\{\cos(\pi J)-1\}$, and $\beta_0^C = -\frac{\beta^{C[NL]}}{2K}\{\cos(\pi K)-1\}$. Here, $y_n$ represents the artificial data generated as follows:

$$y_n \sim \text{Normal}\left(\sum_{i=1}^{I}x_{n,i}^A\beta_i^A + \sum_{j=1}^{J}x_{n,j}^P\beta_j^P + \sum_{k=1}^{K}x_{n,k}^C\beta_k^C, \gamma\right) \qquad n = 1, \ldots, N.$$

 

This paper then sets $I = 10$, $J = 10$, $\gamma = 0.1$ and $N = I \times J \times 10 = 1000$, so that the error terms generated by the normal distributions do not greatly affect the simulation.

In addition, we need to define a bias evaluation function. The estimates of these models can be approximated by using the artificial parameters and the linear components,

$$\widehat{b_i^A} \approx \beta_i^A + s v_i^A, \quad \widehat{b_j^P} \approx \beta_j^P - s v_j^P, \quad \widehat{b_k^C} \approx \beta_k^C + s v_k^C,$$

taking the medians of the estimates as the particular solutions and referring to the general solutions of Eq (4). Here, a small absolute value of $s$ means the model succeeded in recovering the artificial parameters. Thus, this paper defines the bias evaluation function,

$$f(s) = \sum_{i=1}^{I} \left\{ \widehat{b_i^A} - \left( \beta_i^A + s v_i^A \right) \right\}^2$$
$$+ \sum_{j=1}^{J} \left\{ \widehat{b_j^P} - \left( \beta_j^P - s v_j^P \right) \right\}^2 + \sum_{k=1}^{K} \left\{ \widehat{b_k^C} - \left( \beta_k^C + s v_k^C \right) \right\}^2,$$

and calculates $s$ such that the above function satisfies $df/ds = 0$,

$$s = \frac{\sum_{i=1}^{I} v_i^A (\widehat{b_i^A} - \beta_i^A) - \sum_{j=1}^{J} v_j^P (\widehat{b_j^P} - \beta_j^P) + \sum_{k=1}^{K} v_k^C (\widehat{b_k^C} - \beta_k^C)}{\sum_{i=1}^{I} (v_i^A)^2 + \sum_{j=1}^{J} (v_j^P)^2 + \sum_{k=1}^{K} (v_k^C)^2}. \tag{16}$$

## Simulation 1

Simulation 1 focuses on underestimating the linear component of cohort effects under the constraints of shrinking the parameters because the indexes have a large influence on the cohort effects owing to $K = I + J - 1$ in Eq (1). To conduct a systematic simulation, we discuss combinations of the linear components that are the basis of the identification problem. First, this paper assumes three types of slopes for the artificial parameters: 0, +, and –. The total number of combinations here is $3^3 = 27$, since each effect has three patterns. Specifically, $\beta^{A[L]} = 0$ is expressed as (A) 0, $\beta^{P[L]} > 0$ as (P) +, and $\beta^{C[L]} < 0$ as (C) –. In fact, we need only consider $(27 - 1)/2 = 13$ cases since this paper excludes the cases where there is no linear component, such as (A) 0, (P) 0, (C) 0 and where the positive slope is merely reversed to a negative. The combinations are thus cases 1 to 3 having a positive linear component in one factor, cases 4 to 6 having positive linear components in two factors, cases 7 to 9 having positive and negative linear components in two factors, and cases 10 to 13 having linear components in all factors.

Simulation 1 emphasizes the impact of the index weights shown in "Mathematical mechanism" subsection by making the absolute values of the nonlinear components small. This paper sets the variation of the slope to 0.1 and the nonlinear component to 0.05. Specifically, (A) 0 represents $\beta^{A[L]} = 0$ and $\beta^{A[NL]} = 0$, (P) + represents $\beta^{P[L]} = 0.1$ and $\beta^{P[NL]} = 0.05$, and (C) – represents $\beta^{C[L]} = -0.1$ and $\beta^{C[NL]} = -0.05$. To understand the 13 cases, Fig 1 visualizes artificial data that includes only linear components using $y_{i,j} = \beta^{A[L]} v_i^A + \beta^{P[L]} v_j^P + \beta^{C[L]} v_k^C$, and Fig 2 also includes nonlinear components using $y_{i,j} = \beta_i^A + \beta_j^P + \beta_k^C$. The dot plots are visualizations by period and the x-axis represents cohort. The solid lines connect the dots corresponding to $j = 1, 4, 7, 10$ for each period. Here, $y_{i,j}$ in case 1 of Fig 1 increases by 0.1 as the age index increases, and we see the values of older cohorts as higher within the same period. Case

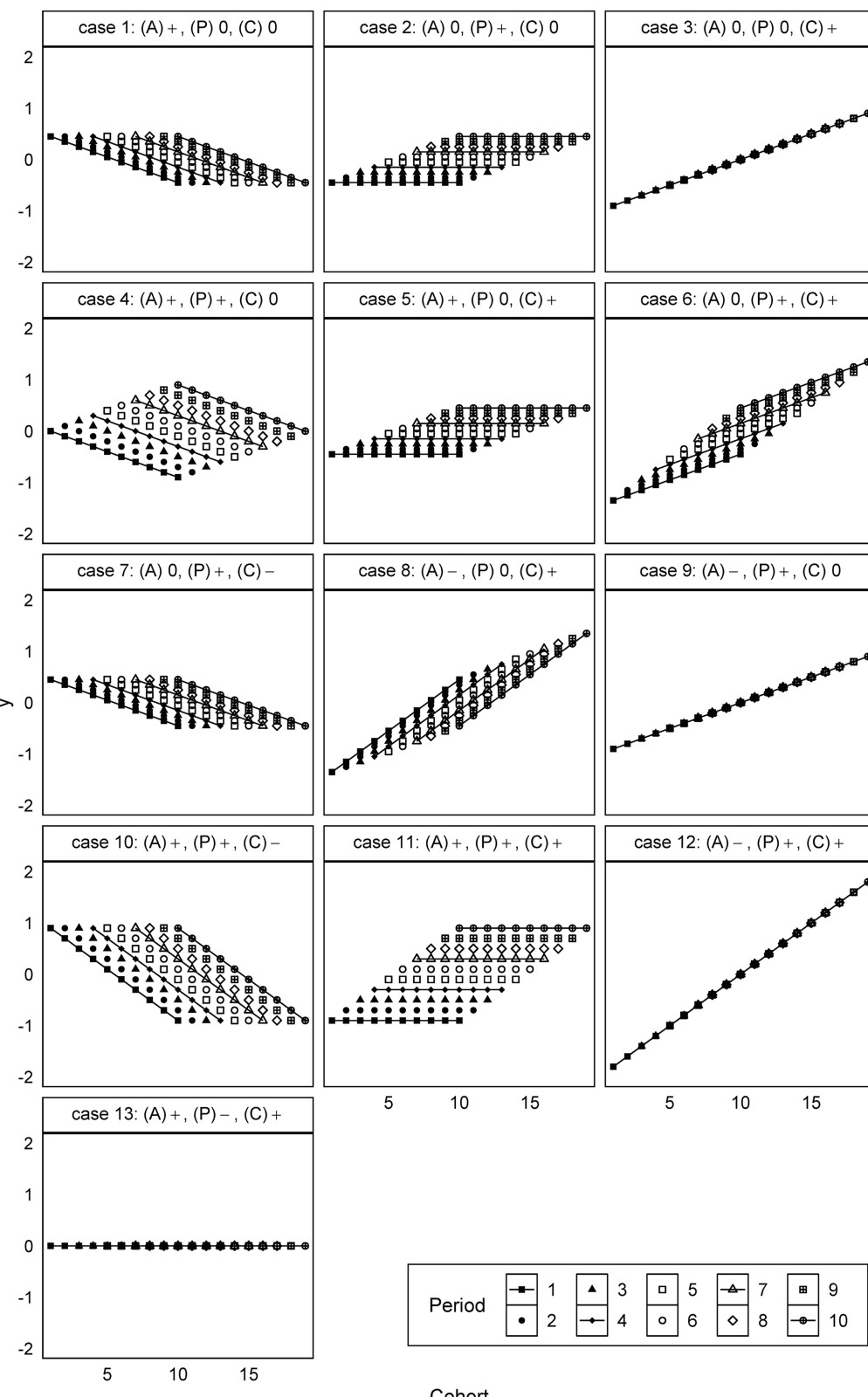

**Fig 1. Artificial data generated with only linear components for age, period, and cohort effects (Simulation 1).**
Note: Each panel represents a different combination of linear slopes as described in Table 2.

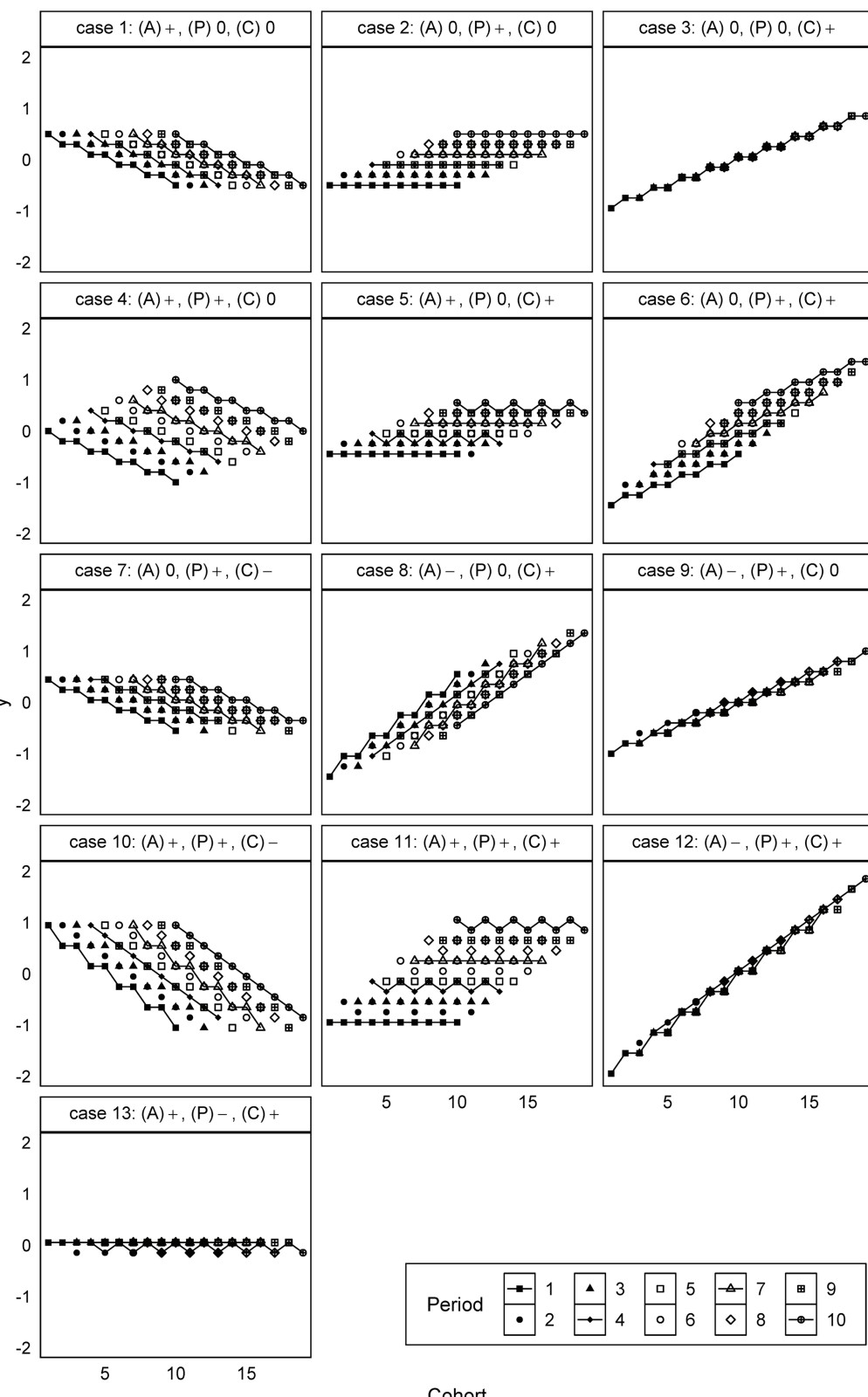

**Fig 2. Artificial data generated with both linear and nonlinear components for age, period, and cohort effects (Simulation 1).** Note: Each panel represents a different combination of linear slopes as described in Table 2.

2 increases by 0.1 as the period index increases and case 3 increases by 0.1 as the cohort index increases. However, Fig 1 shows that cases 1 and 7 are identical and very similar to case 10. Moreover, cases 2 and 5 are identical and very similar to case 11, while cases 3 and 9 are identical and very similar to case 12. In addition, the linear components of case 13 are offset and no variation appears in the artificial data. In other words, the mixture of linear components in the identification problem means that combining different linear components can generate precisely the same data. Unlike Figs 1 and 2 does not reveal identical data. Consequently, this paper verifies whether the models of Bayesian regularization recover the artificial parameters using this small difference.

## Simulation 2

Simulation 2 randomly generates the amounts of change in the linear and nonlinear components according to normal distributions,

$$\beta^{A[L]} \sim \text{Normal}\,(0, 0.1), \quad \beta^{P[L]} \sim \text{Normal}\,(0, 0.1), \quad \beta^{C[L]} \sim \text{Normal}\,(0, 0.1),$$
$$\beta^{A[NL]} \sim \text{Normal}\,(0, 0.1), \quad \beta^{P[NL]} \sim \text{Normal}\,(0, 0.1), \quad \beta^{C[NL]} \sim \text{Normal}\,(0, 0.1).$$

This paper generates artificial data 500 times, obtains $s$ in Eq (16) for each model, and evaluates the models by calculating $\delta$,

$$\delta = \sqrt{\frac{1}{T}\sum_{t=1}^{T} s_t^2}, \tag{17}$$

where $T$ denotes the number of times the model converged in the simulation.

Fig 3 visualizes the linear and nonlinear components of the generated artificial parameters as dots. Here, this paper classifies the artificial parameters into four main patterns: (1) no linear and nonlinear components, (2) only linear components, (3) only nonlinear components, and (4) both linear and nonlinear components. This simulation has an equal probability of the patterns containing only linear components and only nonlinear components.

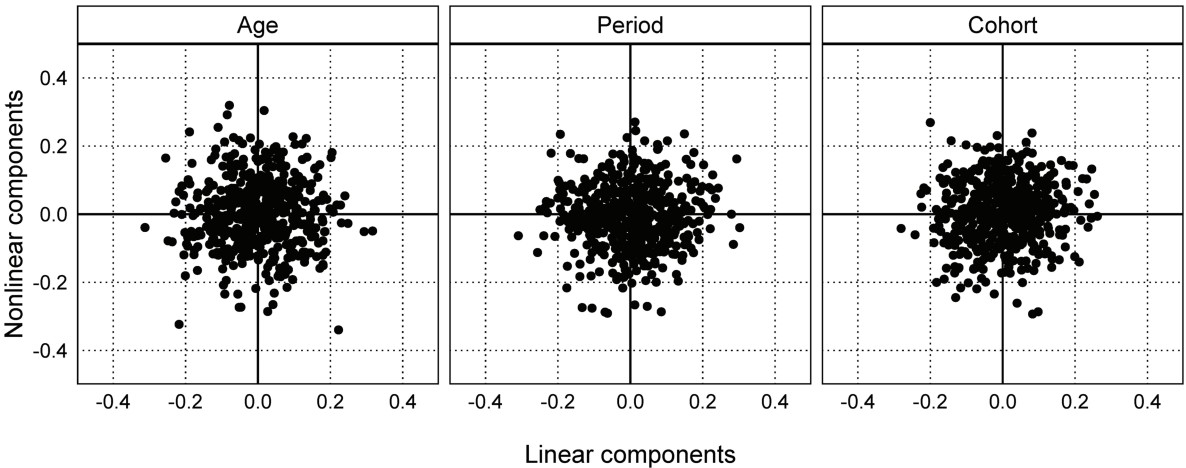

**Fig 3. Linear and nonlinear components of artificial parameters (Simulation 2).**

### Simulation 3

Simulation 3 modifies the assumption of Simulation 2 that the pattern with only linear components and only nonlinear components appear equally. In addition, Simulation 3 generates the artificial parameters by considering Bayesian regularization that the assumption estimates linear components using nonlinear components. In other words, we set the artificial parameters so that the pattern with only linear components is less likely to appear than only nonlinear components. Specifically, this paper randomly generates the amounts of change in the nonlinear components according to normal distributions and uses their absolute values to generate the linear components,

$$\beta^{A[NL]} \sim \text{Normal}\,(0, 0.1), \quad \beta^{A[L]} \sim \text{Normal}\,(0, |\beta^{A[NL]}|),$$
$$\beta^{P[NL]} \sim \text{Normal}\,(0, 0.1), \quad \beta^{P[L]} \sim \text{Normal}\,(0, |\beta^{A[NL]}|),$$
$$\beta^{C[NL]} \sim \text{Normal}\,(0, 0.1), \quad \beta^{C[L]} \sim \text{Normal}\,(0, |\beta^{A[NL]}|).$$

Fig 4 visualizes the linear and nonlinear components of the generated artificial parameters as dots. Moreover, we add to Fig 4 the gray area bounded by $\beta^{[NL]} = 0.2\beta^{[L]}$ and $\beta^{[NL]} = -0.2\beta^{[L]}$, including the horizontal axis. The absence of dots in the gray area indicates that Simulation 3 does not generate the artificial parameter containing only linear components.

### Results

The three models of Bayesian regularization were implemented using the probabilistic programming language Stan [18] and were run in R [19]. Sampling settings were chains = 4, iter = 2000, warmup = 500, and thin = 3. The lower bounds of $\sigma^A$, $\sigma^P$, and $\sigma^C$ in the random effects model were set to 0.05 in order to search for parameters in a wide range, as this model can get stuck in locally optimal solutions. This paper judges the model to have converged when all parameters satisfy $\widehat{R} < 1.05$.

The three models in Simulation 1 satisfy the convergence criterion in all cases. Table 2 summarizes the systematic combinations of the linear components and the results of the three

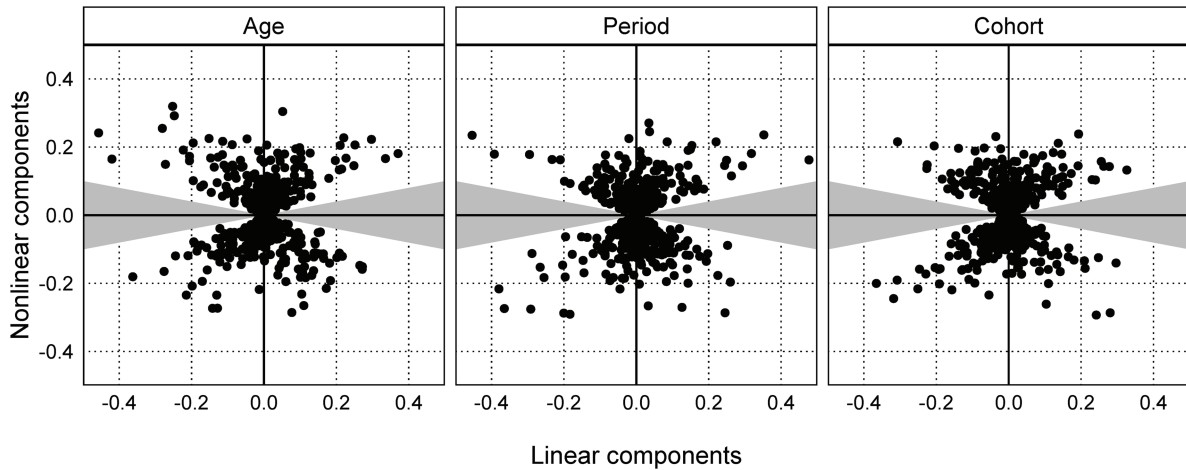

**Fig 4. Linear and nonlinear components of artificial parameters (Simulation 3).**

**Table 2. Combinations of linear components and results (Simulation 1).**

| case | Artifical parameters | | | Random effects model | | Ridge regression model | | Random walk model | |
|---|---|---|---|---|---|---|---|---|---|
| | (A) | (P) | (C) | $s$ | | $s$ | | $s$ | |
| 1 | + | 0 | 0 | 0.000 | A | −0.012 | A | 0.000 | A |
| 2 | 0 | + | 0 | 0.000 | A | 0.012 | A | 0.000 | A |
| 3 | 0 | 0 | + | −0.099 | E | −0.077 | D | 0.000 | A |
| 4 | + | + | 0 | 0.000 | A | 0.000 | A | 0.000 | A |
| 5 | + | 0 | + | −0.100 | E | −0.089 | E | 0.000 | A |
| 6 | 0 | + | + | −0.099 | E | −0.066 | D | −0.001 | A |
| 7 | 0 | + | − | 0.100 | E | 0.089 | E | 0.001 | A |
| 8 | − | 0 | + | −0.099 | E | −0.066 | D | 0.001 | A |
| 9 | − | + | 0 | 0.001 | A | 0.023 | B | 0.000 | A |
| 10 | + | + | − | 0.100 | E | 0.078 | D | 0.088 | E |
| 11 | + | + | + | −0.100 | E | −0.078 | D | −0.092 | E |
| 12 | − | + | + | −0.100 | E | −0.054 | C | −0.020 | B |
| 13 | + | − | + | −0.101 | E | −0.101 | E | −0.105 | E |

Note: For example, $\beta^{A[L]} = 0$ is expressed as (A) 0, $\beta^{P[L]} > 0$ as (P) +, and $\beta^{C[L]} < 0$ as (C) −. This paper calculated $s$ in Eq (16). The letters A through E are used to categorize the results: A if the absolute value of $s$ is less than 0.02, B if less than 0.04, C if less than 0.06, D if less than 0.08, and E if 0.08 or more.

models. The letters A through E that appear in three of the table columns are used to categorize the results: A if the absolute value of $s$ is less than 0.02, B if less than 0.04, C if less than 0.06, D if less than 0.08, and E if 0.08 or more. As shown, among the 13 cases of artificial data, 10 cases in the random walk model rated B or better (i.e., the value of $s$ was less than 0.04) as compared to 4 cases in the random effects and ridge regression models, indicating that the random walk model performed relatively well.

The cases where the models failed to effectively recover the artificial parameters in Simulation 1 contain the linear component of cohort effects. First, constraints shrinking the parameters, such as in Bayesian regularization, always fail case 13, where the linear components completely cancel. Moreover, we found the estimated linear component of the cohort effects in the failed cases to be close to zero, because (C) + leads to $s < 0$ and (C) – leads to $s > 0$. Specifically, Fig 5, which visualizes case 3, shows that the estimated slope of the cohort effects becomes horizontal, and the linear component is incorrectly assigned to the other effects. The random effects and ridge regression models obtain the estimates like the artificial parameters in case 9 because the age effects have a negative slope and the period effects have a positive slope. However, the random walk model did not underestimate the linear component of the cohort effects.

In Simulation 2, the number of times the models converged was 479 for the random effects model, 500 for the ridge regression model, and 487 for the random walk model. In addition, $\delta$ in Eq (17) was 0.092 for the random effects model, 0.077 for the ridge regression model, and 0.088 for the random walk model. Fig 6 shows that the histograms of $s$ in Eq (16) for all three models are widely spread, indicating that none of the models can recover the artificial parameters.

Unlike Simulation 1, the random walk model did not perform well in Simulation 2. The reason is that different artificial parameters generate the same artificial data, as shown in Fig 1. For example, the artificial parameters shown in Fig 7 generate the artificial data of case 3, meaning that the random walk model in this case incorrectly assigns the linear component to the cohort effects that include the nonlinear components. Therefore, it is impossible to decide which model performs well with no constraints on the linear and nonlinear components.

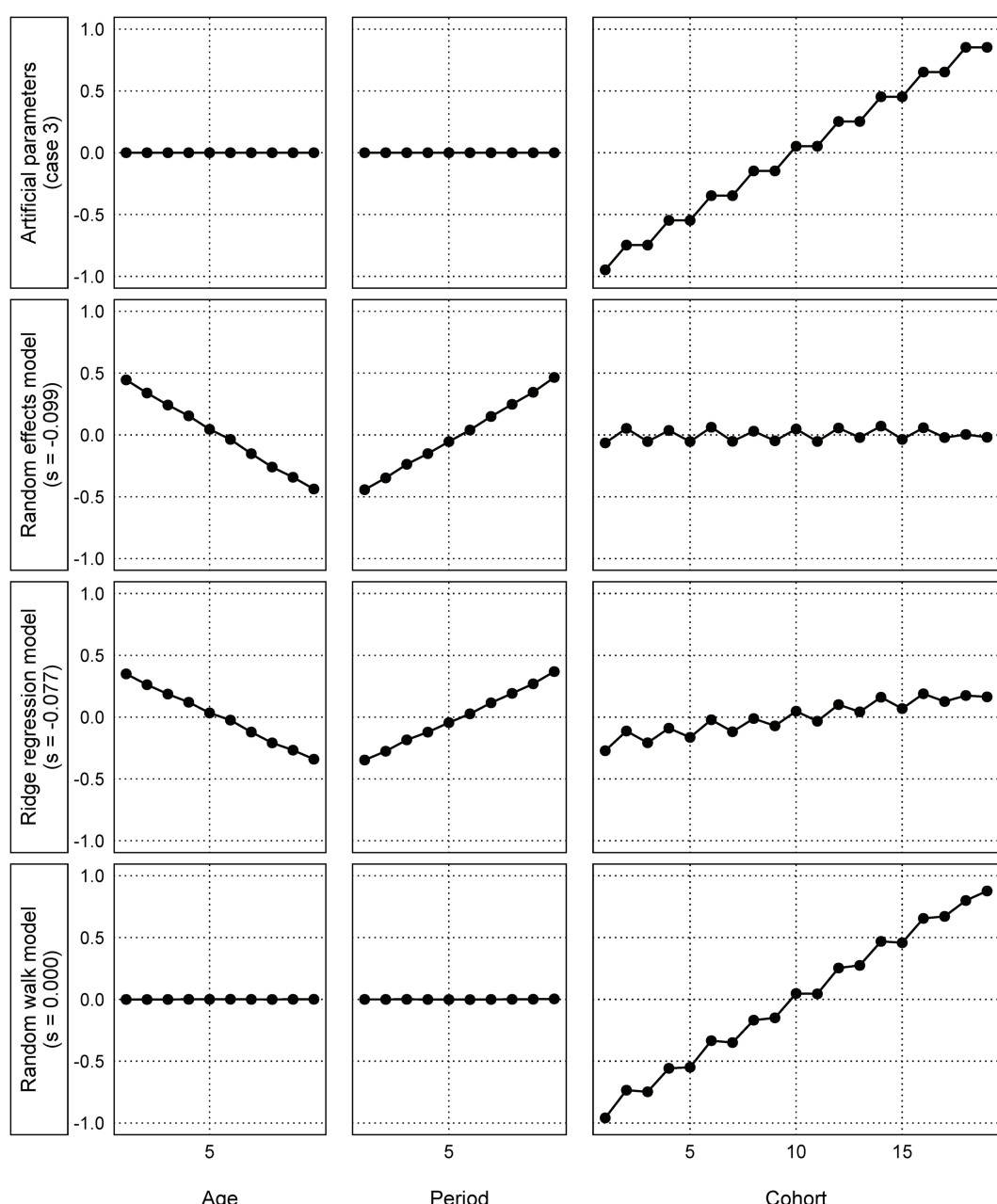

**Fig 5. Comparison of the three models' estimates (case 3 in Simulation 1).**

In Simulation 3, the number of times the models converged was 469 for the random effects model, 500 for the ridge regression model, and 499 for the random walk model. In addition, $\delta$ in Eq (17) was 0.063 for the random effects model, 0.070 for the ridge regression model, and 0.033 for the random walk model. Fig 8 shows that since the histogram of $s$ in Eq (16) for the random walk model is concentrated near zero, this model has less bias than the other models.

The models applying Bayesian regularization estimates the linear components using the nonlinear components and the priors. Simulation 3 is less likely to generate artificial parameters with only linear components, which makes random walk models advantageous because

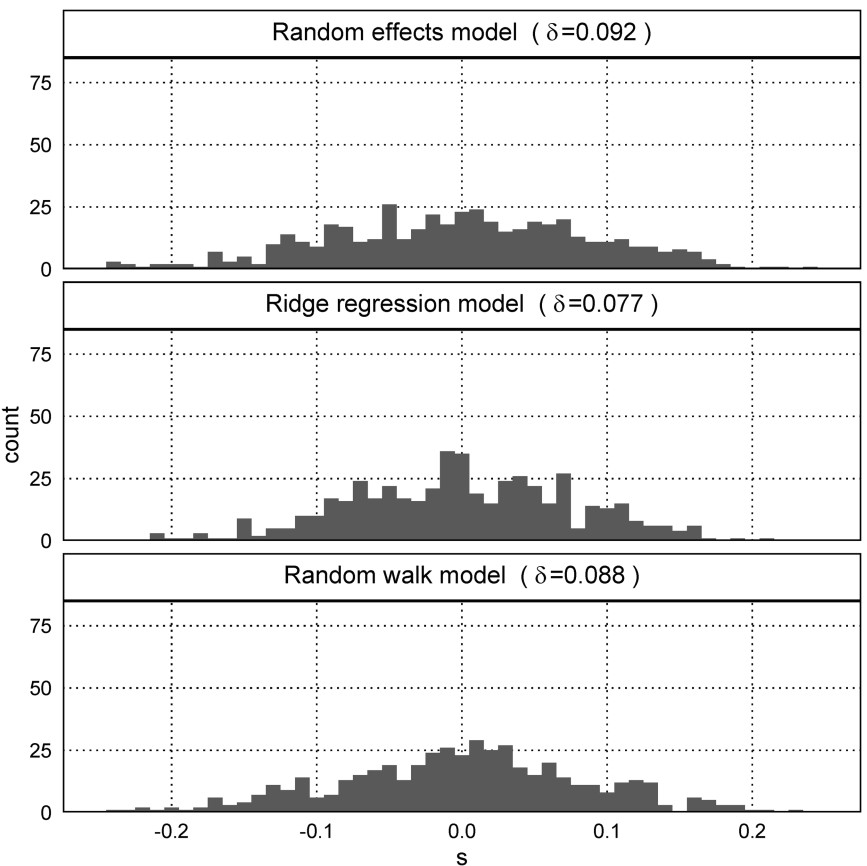

**Fig 6. Histograms of *s* for the three models (Simulation 2).**

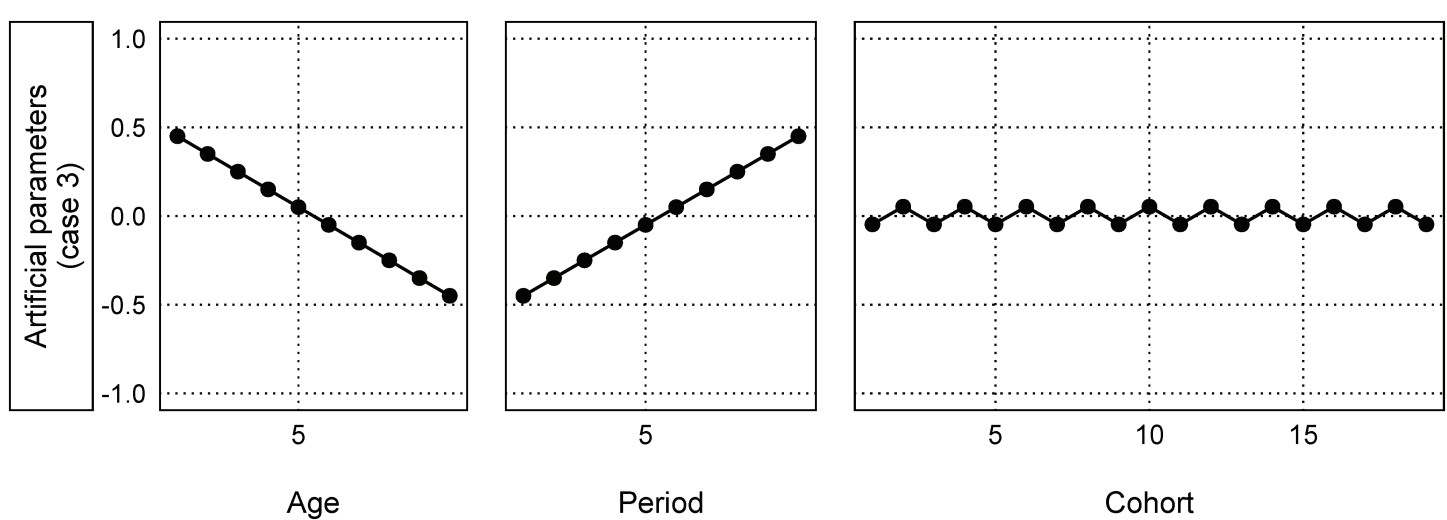

**Fig 7. Another artificial parameters to generate case 3 of Simulation 1.**

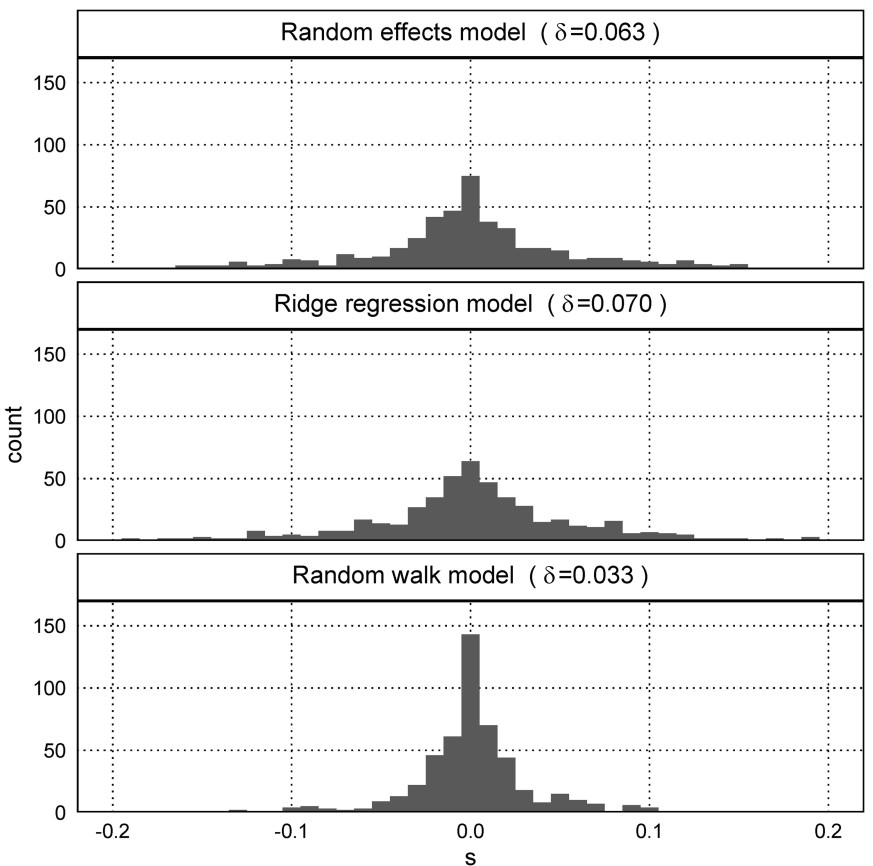

**Fig 8. Histograms of *s* for the three models (Simulation 3).**

this simulation does not generate patterns like Fig 7. Here, the nonlinear components of the three effects determine the lower bounds of $\sigma^A$, $\sigma^P$, and $\sigma^C$ in the log prior probabilities of the random effects model and the random walk model, and their sigmas affect the assignment of the linear components. However, the ridge regression model cannot effectively use the nonlinear components because this model uses $\lambda$ that is common to the three effects. Furthermore, Table 2 shows that the random effects model underestimates the linear components of the cohort effects than the random walk model. Therefore, the random walk model performs relatively well.

## Discussion

### Findings

This paper focused on the three models in APC analysis of Bayesian regularization using the priors of normal distributions. The random effects model refers to multilevel analysis, the ridge regression model is equivalent to the intrinsic estimator, and the random walk model refers to the Bayesian cohort model. We verified some simulation with settings for the linear and nonlinear components. Simulation 1 considers the systematic combinations of the linear components and emphasizes the impact of the indexes by making the absolute values of the nonlinear components small. Simulation 2 randomly generates the amounts of change in the linear and nonlinear components according to normal distributions. Simulation 3

sets the artificial parameters so that the pattern with only linear components is unlikely to appear. Table 3 briefly summarizes the settings for the amount of change in each component of Simulation 1 to 3. The purpose of this paper is to suggest conditions for using the random walk model by comparing the three models through these simulations in terms of how well artificial parameters are recovered.

In general, the constraints of shrinking the parameters tend to drive the linear component of the cohort effects close to zero because the indexes have a large influence on the cohort effects owing to $K = I + J - 1$ in Eq (1). The results of Simulation 1 showed the random effects model reproduced the findings [4,5] that the linear component of the cohort effects becomes flat. Unlike the other two models, the random walk model mitigated underestimating the linear component of the cohort effects and successfully recovered the artificial parameters if one or more of the effects were zero. On the other hand, Simulation 2 showed none of the models can recover the artificial parameters. Therefore, the mitigation of underestimating the linear component of the cohort effects, mentioned in the previous study [11], does not determine which model performs well with no constraints on the linear and nonlinear components. In Simulation 3, as the pattern with only linear components was unlikely to appear, the random walk model had less bias than the other models.

Applying Bayesian regularization in APC analysis is to statistically estimate mathematically indistinguishable linear components by using mathematically identifiable nonlinear components and priors. This shrinking constraint always fails to estimate the cases where the linear components completely cancel each other, such as case 13 in Simulation 1. However, the setting in Simulation 3 is consistent with the assumption of Bayesian regularization using the nonlinear components to estimate the linear components. In addition, this means that the artificial parameters in Fig 7, where the random walk model fails, are less likely to appear. Here, the ridge regression model cannot effectively use nonlinear components to estimate linear components. Furthermore, Table 2 shows that the random effects model underestimates the linear components of the cohort effects than the random walk model. As a result, the random walk model in Simulation 3 recovered the artificial parameters generated by trigonometric functions rather than random walks.

## Applicability

This paper classified artificial parameters into four main patterns: (1) no linear and nonlinear components, (2) only linear components, (3) only nonlinear components, and (4) both linear and nonlinear components. Among them, the simulations using trigonometric functions for the nonlinear components showed that the random walk model recovered artificial parameters better than the other models, if the pattern with only linear components is unlikely to appear. This subsection briefly discusses the possibility that the random walk models can estimate data generating processes other than trigonometric functions. For example, polynomial functions are more common than trigonometric functions in analysis, as there is polynomial regression. Therefore, we describe artificial parameters using them.

**Table 3. Settings for the amount of change in each component (Simulation 1 to 3).**

|  | Linear component | Nonlinear component |
|---|---|---|
| Simulation 1 | Three types: 0, 0.1, −0.1 | One-half of the linear component |
| Simulation 2 | Normal $(0, 0.1)$ | Normal $(0, 0.1)$ |
| Simulation 3 | Normal $(0, |$the nonlinear component$|)$ | Normal $(0, 0.1)$ |

Let $m = 1, \ldots, M$ denote a index and $v_m$ denote a centered index, $v_m = m - \frac{M+1}{2}$. Moreover, $h = 1, \ldots, H$ denotes the exponent of polynomial functions and $z_{m,h}$ denotes the standardized $(v_m)^h$,

$$z_{m,h} = \left\{ (v_m)^h - \frac{1}{M} \sum_{m=1}^{M} (v_m)^h \right\} \bigg/ \sqrt{\frac{1}{M} \sum_{m=1}^{M} \left\{ (v_m)^h - \frac{1}{M} \sum_{m=1}^{M} (v_m)^h \right\}^2}.$$

Using $z_{m,h}$, $\eta_m$ is a centered polynomial function,

$$\eta_m = \sum_{h=1}^{H} w_h z_{m,h} - \frac{1}{M} \sum_{m=1}^{M} \sum_{h=1}^{H} w_h z_{m,h},$$

where $w_h$ denotes a random number generated by a normal distribution,

$$w_h \sim \text{Normal}(0, 0.1) \qquad h = 1, \ldots, H.$$

Here, $\eta^{[L]}$ denotes a linear component of the polynomial function, $\eta^{[NL]}$ denotes a non-linear component, and $SD^{[NL]}$ denotes a standard deviation of the nonlinear components. The above represents $\eta_m = \eta^{[L]} v_m + \eta_m^{[NL]}$, where $\eta^{[L]}$ minimizes $\sum_{m=1}^{M} (\eta_m^{[NL]})^2$. Specifically, we obtain the following with reference to matrix calculations,

$$\eta^{[L]} = \left\{ \sum_{m=1}^{M} (v_m)^2 \right\}^{-1} \left( \sum_{m=1}^{M} v_m \eta_m \right),$$

and the standard deviation is

$$SD^{[NL]} = \sqrt{\frac{1}{M} \sum_{m=1}^{M} (\eta_m^{[NL]})^2}.$$

Fig 9 visualizes $\eta^{[L]}$ and $SD^{[NL]}$ of the generated artificial parameters as dots. Moreover, we add to Fig 9 the gray area bounded by $SD^{[NL]} = 0.5\eta^{[L]}$ and $SD^{[NL]} = -0.5\eta^{[L]}$, including the horizontal axis. As in Simulation 3, the absence of dots in the gray area indicates that the polynomial function does not generate the artificial parameter containing only linear components. Fig 9 shows that there are many dots in the gray area at $H = 2$ and no dots at $H = 8$. The reason is that $w_1$ affects only the linear component, while $w_3$ and $w_5$ affect the linear and nonlinear components. In addition, even if the absolute value of $w_3$ is large, there is the possibility that the linear component is close to zero due to $w_1$ and $w_5$. However, it is difficult to cancel out the nonlinear component of $w_3 z_{m,3}$ with other terms. In summary, the polynomial functions in this subsection are less likely to generate the pattern with only linear components as the degree of the polynomial increases. Therefore, the random walk model may have a smaller bias than the other models even when data generating processes can be approximated by polynomial functions with a large degree of polynomial.

Finally, this paper has several limitations, such as only discussing the main three models applying Bayesian regularization, not changing the indexes, for example $I = 12$ or $J = 8$, and not considering nonlinear components other than trigonometric functions. We should verify other data generating processes, and the simulations and bias evaluation functions in this paper will be useful in such cases. This paper showed the simulations not only where no model can recover the artificial parameters but also where the random walk model performs

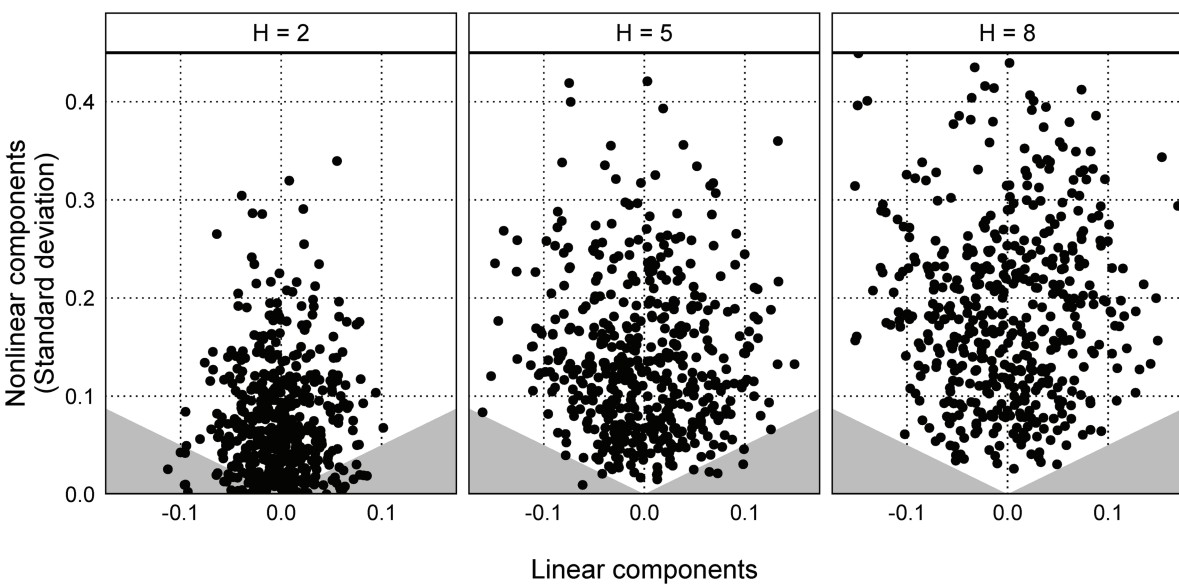

**Fig 9. Linear and nonlinear components generated by polynomial functions.**

well. Therefore, future studies need to investigate not only failure cases but also constraints that reduce bias by imposing weak conditions on linear and nonlinear components.

## Supporting information

**S1 Fig. Comparison of the three models' estimates.**
(PDF)

**S1 Appendix. Stan codes to implement Bayesian regularization models.**
(PDF)

**S2 Appendix. R codes to reproduce the systematic simulation.**
(PDF)

## Author contributions

**Conceptualization:** Yuta Matsumoto.

**Formal analysis:** Yuta Matsumoto.

**Investigation:** Yuta Matsumoto.

**Methodology:** Yuta Matsumoto.

**Software:** Yuta Matsumoto.

**Validation:** Yuta Matsumoto.

**Visualization:** Yuta Matsumoto.

**Writing – original draft:** Yuta Matsumoto.

**Writing – review & editing:** Yuta Matsumoto.

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
