## [Decision Letter · Decision Letter 0]

17 Mar 2025

PONE-D-24-51931Some simulations of age-period-cohort analysis: Conditions for using random walk modelPLOS ONE

Dear Dr.  Matsumoto ,

Thank you for submitting your manuscript to PLOS ONE. After careful consideration, we feel that it has merit but does not fully meet PLOS ONE’s publication criteria as it currently stands. Therefore, we invite you to submit a revised version of the manuscript that addresses the points raised during the review process.

We look forward to receiving your revised manuscript.

Kind regards,

Md. Kamrujjaman, Ph.D

Academic Editor

PLOS ONE

Journal Requirements:

2. We note that your Data Availability Statement is currently as follows: All relevant data are within the manuscript and in Supporting Information files.

Reviewers' comments:

Reviewer's Responses to Questions

**Comments to the Author**

1. Is the manuscript technically sound, and do the data support the conclusions?

Reviewer #1: Partly

Reviewer #2: Partly

2. Has the statistical analysis been performed appropriately and rigorously? 

Reviewer #1: Yes

Reviewer #2: Yes

3. Have the authors made all data underlying the findings in their manuscript fully available?

Reviewer #1: Yes

Reviewer #2: Yes

4. Is the manuscript presented in an intelligible fashion and written in standard English?

Reviewer #1: Yes

Reviewer #2: Yes

5. Review Comments to the Author

Reviewer #1: 1. General

The manuscript investigates the identification problem in age-period-cohort (APC) analysis, a common issue in time-series analysis across various fields. The author focuses on three Bayesian models used to address this problem: the random effects model, the ridge regression model, and the random walk model. Across three simulations using artificial data, the paper aims to determine the conditions under which the random walk model is preferable, specifically in context to its ability to mitigate the underestimation of the linear component of cohort effects.

The manuscript has several key strengths, including its focus on the highly relevant and well-recognised identification problem inherent in APC analysis. The methodology is clearly articulated, with the simulation design and the specifications of the models described in sufficient detail. The author's approach of using simulations to compare the performance of different models is appropriate and valuable. Finally, the manuscript benefits from a well-structured and logical organisation, enhancing its overall readability and accessibility.

However, the manuscript’s ability to generalise the results may be limited due to the specific parameters used in simulations. The analysis and the specific recommendations for model selection could be strengthened. The author needs to clarify the limitations of the study, refine the conclusions, and provide a more nuanced discussion of the role of a priori information. I recommend that the manuscript be considered for publication after addressing these significant revisions.

2. Abstract

The abstract adequately summarises the central research question, the methodological approach, and the primary findings of the study. However, its impact could be strengthened by including a more specific description of the conditions under which the random walk model demonstrates “superior” performance.

Further, this section in particular could benefit from several improvements in clarity and precision. Certain phrases, such as the description of how multilevel analysis impacts cohort effects, are awkwardly phrased and could be reworded for smoother flow. Vague language, particularly regarding the "various patterns" of simulations used in previous studies, should be clarified with more specific details. Furthermore, breaking down some of the longer sentences would improve overall readability and ensure the abstract's key messages are easily grasped by the reader.

For example, "Applying multilevel analysis results in the linear component of the cohort effects close to zero and the intrinsic estimator is sensitive to the coding of the design matrix" is a bit awkward and could be rephrased for better flow.

3. Introduction

The introduction effectively establishes the context and rationale for the study. It clearly defines the APC identification problem, highlighting its significance in time-series analysis. The limitations of current approaches are succinctly explained, adequately justifying the need for the comparative analysis presented.

4. Theory

The Theory section is a strong point of the manuscript, providing a solid foundation for understanding the complexities of APC analysis and the proposed solutions. The section then clearly delineates the three Bayesian models under consideration outlining their mathematical formulations and underlying assumptions. Particularly valuable is the detailed discussion of linear and non-linear components and the impact of index weights. This section provides useful insights into the mechanisms driving the differences in model performance, particularly highlighting why the random walk model might mitigate the underestimation of the linear component of cohort effects under specified simulated conditions.

Line 170-172: "However, Sakaguchi and Nakamura [11] suggested that the random walk model performs well compared to the random effects model; consequently, this paper examines the impact of the index weights." The phrase "performs well" is vague.

5. Methods

The Methods section provides a clear description of the simulation design. The rationale behind the chosen simulation parameters and the overall approach is well-justified. The decision to use trigonometric functions for generating non-linear components is clearly explained, and while acknowledging potential limitations in generalisability, the author provides a reasonable justification for this choice within the context of the study. Furthermore, the bias evaluation function, based on the parameter 's', is appropriate for assessing the models' ability to recover the true underlying parameters and provides a quantifiable metric for comparison. However, explicitly acknowledging that different data-generating mechanisms might yield different results, and briefly explaining why trigonometric functions were chosen over other potential functions, could enhance this section.

6. Results

The Results section effectively communicates the findings of each simulation, employing a clear and organised presentation style. The interpretation of the results is generally sound and logically follows from the data presented. However, the discussion of the results could be improved by more strongly emphasising the limitations of the study, particularly the potential for limited generalisability due to the specific parameter settings used in the simulations. While the results offer valuable insights, a more explicit acknowledgement of these limitations would provide a more balanced and nuanced interpretation of the findings - especially for applied researchers.

7. Discussion

The Discussion section summarises the main findings of the study and begins to explore their implications for APC analysis. It connects the results back to the research questions and provides an initial interpretation of their significance. However, the discussion needs to be significantly expanded in several key areas. Specifically, the limitations of the study, particularly the potential for limited generalisability due to the specific parameters used in the simulation, deserve a more thorough treatment. For example, are there specific types of data or research questions where these limitations are particularly problematic? Further, while acknowledging that the random walk model “performed well” under the simulated conditions, the discussion should provide a more nuanced perspective on the applicability of these findings to real-world data (and possibly suggest avenues for future research to further investigate these limitations and enhance external validity). Also, the role of a priori information and its influence on model selection should be discussed in greater depth, for example, addressing how it might affect the performance of each model.

Line 378-380: "For the above reason, this paper expects the random walk model to perform relatively better than the other two models in real data if there is no other strong a priori information [20]." This conclusion is too broad. The simulations demonstrate the random walk model's advantage under specific conditions, but more evidence is needed to support its general superiority in the absence of strong a priori information. The author should explicitly state that the conclusion applies to scenarios similar to Simulation 3, where only linear components are unlikely. A more cautious conclusion would focus on the specific circumstances where the random walk model demonstrated an advantage in the simulations. It is more accurate to conclude that the random walk model is advantageous when there are both linear and non-linear components for each age, period, and cohort.

Line 381-383: "Finally, this paper has several limitations, such as only discussing the main three models applying Bayesian regularisation, not changing the indexes, for example I = 12 or J = 8, and not considering nonlinear components other than trigonometric functions." This accurately describes the limitations of the paper. The author should therefore expand the discussion of these limitations in the discussion section.

8. Tables & Figures

The figures and tables are, for the most part, well-constructed and contribute to the clarity and interpretability of the results. The tables provide a concise and well-organised summary of the simulation results, facilitating a quick comparison across models and conditions. However, the effectiveness of some figures could be enhanced by providing more descriptive captions. Currently, the captions are brief and could benefit from additional details to make them self-explanatory.

For example, instead of "Artificial data with only linear components (Simulation 1)", it could be "Simulation 1: Artificial data generated with only linear components for age, period, and cohort effects. Each panel represents a different combination of linear slopes as described in Table 2."

Reviewer #2: Review of the paper with manuscript ID: PONE-D-24-51931

"Some simulations of age-period-cohort analysis: Conditions for using random walk

model"

The manuscript addresses an important issue in age-period-cohort analysis, specifically the identification problem and how different Bayesian regularization methods handle it. The concerns and suggestions of reviewers have been partially implemented. The title of paper in now acceptable and more it is more relevant to the topic of the paper than the previous title. I think some parts still need to be revised to become a standard paper.

Comments to authors:

1) The paper is well-structured, providing a clear breakdown of the theory, methods, and results. However, the writing can be improved for better readability. Some sentences, particularly in the introduction and discussion, are long and complex, making comprehension difficult. Simplifying them would enhance clarity.

2) Consider adding a brief overview at the end of the introduction to explicitly state the key findings. This will help guide the reader through the study.

3) The study compares three Bayesian regularization models (random effects, ridge regression, and random walk) in the context of APC analysis. While the motivation for the study is clear, the justification for the choice of trigonometric functions for nonlinear components could be elaborated. Why were these functions selected over other nonlinear patterns?

4) The definition of bias evaluation function is useful, but it would be beneficial to discuss any potential limitations or assumptions associated with its use.

5) The study presents three simulations to analyze model performance.

• Simulation 1 effectively illustrates the impact of index weights on underestimation of cohort effects.

• Simulation 2 highlights the challenges in parameter recovery but does not fully explore why none of the models performed well. Could additional constraints or adjustments improve the performance?

• Simulation 3 shows that the random walk model performs better when only linear components are unlikely to appear. However, the assumption that such patterns are less likely in real data should be supported with references or empirical examples.

6) The conclusion suggests that the random walk model performs better under certain conditions. It would be helpful to specify the types of datasets or research questions where this model would be most applicable.

6. PLOS authors have the option to publish the peer review history of their article (what does this mean?). If published, this will include your full peer review and any attached files.

Reviewer #1: No

Reviewer #2: No

---

## [Author Response · Author response to Decision Letter 1]

25 Apr 2025

Response to reviewers is attached as PDF file.

---

## [Decision Letter · Decision Letter 1]

26 Jun 2025

PONE-D-24-51931R1Some simulations of age-period-cohort analysis applying bayesian regularization: Conditions for using random walk modelPLOS ONE

Dear Dr. Matsumoto,

Thank you for submitting your manuscript to PLOS ONE. After careful consideration, we feel that it has merit but does not fully meet PLOS ONE’s publication criteria as it currently stands. Therefore, we invite you to submit a revised version of the manuscript that addresses the points raised during the review process.

We look forward to receiving your revised manuscript.

Kind regards,

Md. Kamrujjaman, Ph.D

Academic Editor

PLOS ONE

Journal Requirements:

Reviewers' comments:

Reviewer's Responses to Questions

**Comments to the Author**

1. If the authors have adequately addressed your comments raised in a previous round of review and you feel that this manuscript is now acceptable for publication, you may indicate that here to bypass the “Comments to the Author” section, enter your conflict of interest statement in the “Confidential to Editor” section, and submit your "Accept" recommendation.

Reviewer #1: All comments have been addressed

Reviewer #3: All comments have been addressed

2. Is the manuscript technically sound, and do the data support the conclusions?

Reviewer #1: Yes

Reviewer #3: Yes

3. Has the statistical analysis been performed appropriately and rigorously? 

Reviewer #1: Yes

Reviewer #3: Yes

4. Have the authors made all data underlying the findings in their manuscript fully available?

Reviewer #1: Yes

Reviewer #3: Yes

5. Is the manuscript presented in an intelligible fashion and written in standard English?

Reviewer #1: Yes

Reviewer #3: Yes

6. Review Comments to the Author

Reviewer #1: The author has attended to all the reccommendations appropriately. The author has made an effort to address the "cautious conclusion" and "generalisability" points by modifying the wording in the conclusion and adding a discussion on polynomial function. Furthermore, instances of ambiguity and vague language have been clarified. The addition of the section on “Applicability” has strengthened the paper.

Reviewer #3: The author has addressed all review comments from the first round. I suggest one minor revision: including visualizations of a toy dataset to help readers better understand the type of data being considered in this paper.

7. PLOS authors have the option to publish the peer review history of their article (what does this mean?). If published, this will include your full peer review and any attached files.

Reviewer #1: No

Reviewer #3: No

---

## [Author Response · Author response to Decision Letter 2]

2 Jul 2025

I attached the response to the comments as the pdf file.

---

## [Decision Letter · Decision Letter 2]

14 Jul 2025

Some simulations of age-period-cohort analysis applying bayesian regularization: Conditions for using random walk model

PONE-D-24-51931R2

Dear Dr. Matsumoto,

We’re pleased to inform you that your manuscript has been judged scientifically suitable for publication and will be formally accepted for publication once it meets all outstanding technical requirements.

Kind regards,

Md. Kamrujjaman, Ph.D

Academic Editor

PLOS ONE

Additional Editor Comments (optional):

Reviewers' comments:

Reviewer's Responses to Questions

**Comments to the Author**

1. If the authors have adequately addressed your comments raised in a previous round of review and you feel that this manuscript is now acceptable for publication, you may indicate that here to bypass the “Comments to the Author” section, enter your conflict of interest statement in the “Confidential to Editor” section, and submit your "Accept" recommendation.

Reviewer #3: All comments have been addressed

2. Is the manuscript technically sound, and do the data support the conclusions?

Reviewer #3: Yes

3. Has the statistical analysis been performed appropriately and rigorously? 

Reviewer #3: Yes

4. Have the authors made all data underlying the findings in their manuscript fully available?

Reviewer #3: Yes

5. Is the manuscript presented in an intelligible fashion and written in standard English?

Reviewer #3: Yes

6. Review Comments to the Author

Reviewer #3: (No Response)

7. PLOS authors have the option to publish the peer review history of their article (what does this mean?). If published, this will include your full peer review and any attached files.

Reviewer #3: No
